# Self-Prompt SAM: Automatic Prompt SAM Adaptation for Medical Image Segmentation

## Abstract

The Segment Anything Model (SAM), a prompt-driven foundation model for natural image segmentation, has demonstrated impressive zero-shot performance and brought a range of unexplored capabilities to natural image segmentation tasks. However, as a very important branch of image segmentation, the performance of SAM remains uncertain when it is applied to medical image segmentation due to the significant differences between natural images and medical images. Meanwhile, it is harsh to meet the requirements of extra prompts provided, such as points or boxes to specify medical regions, since medical knowledge is not expected from users. In this paper, we aim to adapt pre-trained SAM models worked on from 2D natural images to 3D medical images without any prompts provided. Through the analysis of SAM models, we propose a novel self-prompt SAM adaptation framework for medical image segmentation, named Self-Prompt-SAM. We designed a multi-scale prompt generator combined with the image encoder in SAM to generate auxiliary masks. Then, we use the auxiliary masks to generate bounding boxes as box prompts and utilize Distance Transform to select the points farthest from the boundary as point prompts. Meanwhile, we designed a 3D depth-fused adapter (DfusedAdapter) and injected the DFusedAdapter into each transformer block in the image encoder and mask decoder to enable pre-trained 2D SAM models to extract 3D information and adapt to 3D medical images. Extensive experiments demonstrate that our method outperforms existing state-of-the-art approaches on two challenging public ACDC Bernard et al. (2018) and Synapse Landman et al. (2015) datasets.

## 1 Introduction

The purpose of medical image segmentation is to utilize medical images to segment specific anatomical structures including organs, lesions, and tissues, which can assist in many clinical applications, such as disease diagnosis, surgical planning, and monitoring of disease progression. Deep learning methods Ronneberger et al. (2015); Akkus et al. (2017); Avendi et al. (2016) have achieved numerous and remarkable progress in the field of medical image segmentation for the past few years. However, existing deep learning models are often tailored, which have a strong inductive bias and limit their capacity, for specific tasks.

The rise of foundation models that are trained on vast and diverse datasets has revolutionized artificial intelligence. Benefiting from their remarkable zero-shot and few-shot generalization abilities, a wide range of downstream tasks that adapt a pre-trained model to specific tasks achieve numerous and remarkable progress, not like the traditional methods of training task-specific models from scratch. Recently, the Segment Anything Model (SAM) Kirillov et al. (2023), pre-trained over 1 billion masks on 11 million natural images, has been proposed as a visual foundation model for prompt-driven image segmentation and has gained huge attention due to its impressive zero-shot performance for generating accurate object masks in a fully automatic or interactive way. Based on its strong capabilities in natural image segmentation, can SAM still maintain strong performance when it is applied to medical image segmentation, though there are significant differences between natural images and medical images?

It is infeasible to apply directly. The first reason is SAM needs extra prompts when segmenting specific regions. It is impossible to expect all users to have medical knowledge to provide points in

Figure 1: Experiments to prove which combination of points, boxes, and masks is the best prompt.

specific regions or frame out specific regions. Figure 1(a) shows different prompts, such as points, boxes, and masks. When a user without any medical background is given a raw image, it is hard to identify every class of organs, such as the liver (purple region). Therefore, how to solve the harsh requirements is the first key issue when applying SAM to medical image segmentation. We found the image encoder employs the Vision Transformer (ViT) Dosovitskiy et al. (2020) pre-trained with masked auto-encoder He et al. (2022) as the backbone. Benefiting from ViT's strong representation capabilities, the image encoder extracts essential features of the images with a series of transformer blocks. Therefore, we designed a multi-scale prompt generator (MSPGenerator) combined with the image encoder to generate auxiliary masks. Multiple levels of feature maps are extracted from the image encoder as the input of our MSPGenerator, which is built via a hierarchical structure with convolutional layers to learn different levels and different scale information by gradually upsampling the spatial dimensions to the same size as the ground truth. Finally, auxiliary masks would be utilized for prompts. However, the original SAM segments all possible objects and does not classify each object belonging to which class, which is more like for instance segmentation methods. Our expectation of SAM is to predict each class given raw images as input, like traditional medical image segmentation. To achieve this purpose, our designed MSPGenerator would generate an auxiliary mask for each class. Then we utilize the auxiliary mask to generate a point and a bounding box as the prompt for each class.

The second reason is that directly applying SAM to medical image segmentation tasks does not always obtain good performance when proper prompts are provided. Many works Deng et al. (2023); Hu & Li (2023); Zhou et al. (2023); Mohapatra et al. (2023); Roy et al. (2023); Wang et al. (2023); He et al. (2023) have demonstrated that SAM is imperfect or even fails when some situations occur, such as weak boundaries, low-contrast, and smaller and irregular shape, which is consistent with other investigations Ji et al. (2023a;b). Figure 1(b) illustrates results by different prompts. Even if the prompts are generated by the ground truth, the results are very bad, especially by point prompts. Despite this, we found the most robust way to prompt is by bounding boxes with proper points that should select the point farthest from the boundary of each object, which means the point is as central as possible shown in Figure 3(a). Therefore, we adopt the Euclidean distance transform to calculate the distance from boundaries and obtain the candidate points based on auxiliary masks.

Therefore, adapting SAM to medical image segmentation tasks is the main direction by modifying the structure of SAM. How to appropriately modify the structure has become the most important issue. We not only maximize the utilization of the capabilities of SAM but also adapt SAM to medical image segmentation, which is a trade-off. The best way is to keep all structures, freeze all weights, and only add blocks into SAM to adapt. In this way, we retain the zero-shot capabilities of SAM and adapt SAM to medical image segmentation.

Therefore, we designed several blocks to do adaptation besides the MSPGenerator as follows. **(i)** We adopt the lightweight adapter Houlsby et al. (2019) which is a bottleneck architecture that consists of two fully connected (FC) layers and an activation layer in the middle to modify and inject into each transformer block during finetuning. Unlike classic 2D natural images, many medical scans are 3D volumes such as MRI and CT. To learn extra depth information, we designed a depth fused adapter (DFusedAdapter) that added an invert-bottleneck architecture that consists of two FC layers and an activation layer processing on depth dimension in the middle of the original adapter with a skip connection. In this way, we utilize FC layers to learn depth information to achieve 3D spatial fused adaption. **(ii)** To better learn depth information, we add one learnable depth positional embedding with original positional embeddings in the image encoder and mask decoder respectively. **(iii)** Since the natural images have 3 channels for RGB as the input of SAM and medical images have

varied modalities as channels, we designed an invert-bottleneck modalities adapter (MAdapter) that consists of two convolutional layers to adapt the varied modalities to 3 channels, learn short-term dependencies, and learn the differences between medical images and natural images. **(iv)** The output of SAM is binary segmentation for each object. We generate a set of auxiliary prompts for each class, therefore, SAM would generate a binary segmentation for each class. Through observations, we found the distribution of the output for each binary segmentation is not consistent. However, we expect each pixel belonging to a specific class, it would produce large errors if we directly apply a softmax function for the outputs of all classes. Therefore, we also designed an invert-bottleneck multi-classes adapter (MC=Adapter) that consists of two convolutional layers to adapt binary segmentation to multi-class segmentation.

To summarize, our contributions to this paper are:

- We propose a novel self-prompt SAM (Self-Prompt-SAM) framework for medical image segmentation. To the best of our knowledge, the proposed Self-Prompt-SAM is the first SAM-based image segmentation framework without any prompts provided;
- We propose a novel multi-scale hierarchical prompt generator (MSPGenerator) that utilizes multiple levels of feature maps from the image encoder to generate auxiliary masks for prompts. Through massive experiments, we found the best prompt way is to combine bounding boxes, points (use Euclidean distance transform to generate candidate points), and masks;
- We designed a depth fused adapter (DFusedAdapter) that inserts an invert-bottleneck architecture that consists of two FC layers and an activation layer processing on depth dimension in the middle of the original adapter with a skip connection to learn extra depth information.
- We conduct extensive experiments on two challenging ACDC Bernard et al. (2018) and Synapse Landman et al. (2015) datasets. The results demonstrate that Self-Prompt-SAM achieves state-of-the-art performance. The source code and pre-trained models will be made publicly available.

## 2 RELATED WORK

### 2.1 DEEP LEARNING METHODS FOR MEDICAL IMAGE SEGMENTATION

Convolutional Neural Networks (CNNs), especially an encoder-decoder network U-Net Ronneberger et al. (2015) and its variants Zhou et al. (2018); Milletari et al. (2016); Çiçek et al. (2016), have been demonstrated to achieve excellent performance and play an important role in medical image segmentation. The nnUNet Isensee et al. (2019) is a self-adapting framework for U-Net-based medical image segmentation. Although it only utilizes the basis of 2D and 3D vanilla U-Nets, it involves many tricks for preprocessing, analysis of the attributes of datasets, setting up excellent training strategies, and postprocessing. Particularly, it analyzes datasets to generate an exact architecture and strategies for data augmentation. In this way, it achieves great performance in many segmentation tasks. Therefore, our model utilizes the nnUNet as our codebase combined with SAM. Recently, most works employ a hybrid architecture of convolution and self-attention, which has achieved remarkable progress. TransUNet Chen et al. (2021) first time applied transformer to improve the segmentation results of medical images. SwinUNet Cao et al. (2021) builds a U-shape transformer-based segmentation model on top of transformer blocks. nnFormer Zhou et al. (2021) builds a 3D U-shape segmentation model with designed different transformer blocks to achieve different purposes. All methods are often tailored and trained from scratch.

### 2.2 FOUNDATION MODELS AND PARAMETER-EFFICIENT FINETUNING

Foundation models Brown et al. (2020); OpenAI (2023) aim at developing large-scale, general-purpose language and vision models. And most modern vision models follow the pre-training and fine-tuning paradigm He et al. (2019); Hu et al. (2021). Recently, the Segment Anything Model (SAM) Kirillov et al. (2023), pre-trained over 1 billion masks on 11 million natural images, has been proposed as a visual foundation model for prompt-driven image segmentation and has gained huge attention. The goal of parameter-efficient finetuning Houlsby et al. (2019); Pan et al. (2022); Hu et al. (2021) is to decrease the number of trainable parameters and reduce the computation cost while achieving or surpassing the performance of full finetuning. AIM Yang et al. (2023) designed spatial adaptation, temporal adaptation, and joint adaptation to equip a foundation model with spatiotemporal reasoning capability, which adapted the 2D foundation model to 3D

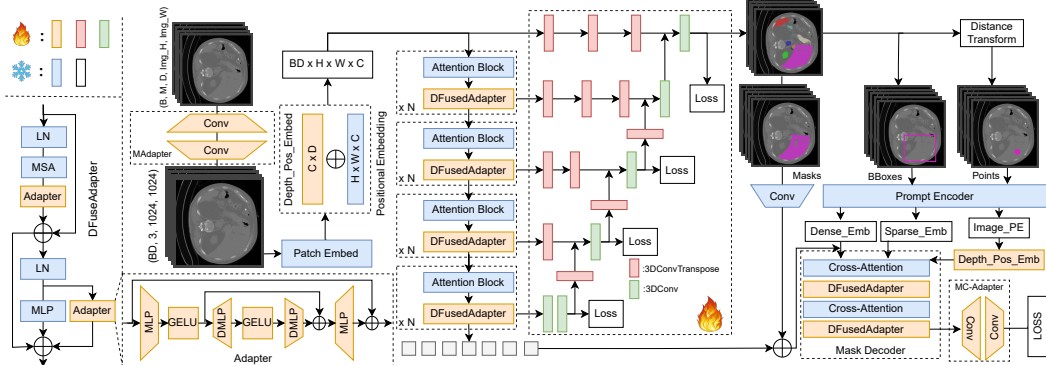

Figure 2: The overview architecture of proposed Self-Prompt-SAM.

video action recognition tasks. In this paper, we utilize the 2D SAM with designed adapters to equip spatiotemporal reasoning capability for 3D medical image segmentation.

## 2.3 SAM-BASED MEDICAL IMAGE SEGMENTATION

SAM-based medical image segmentation works can be divided into two main categories. The first line of works Deng et al. (2023); Hu & Li (2023); Zhou et al. (2023); Mohapatra et al. (2023); Roy et al. (2023); Wang et al. (2023); He et al. (2023) aim to evaluate the performance of SAM in different medical image segmentation tasks with different modes by manual prompt provided. In summary, the evaluation works on different datasets show that the performance of SAM varies significantly across different datasets when directly applied to medical image segmentation. SAM achieves remarkable performance in some specific objects and modalities compared to state-of-the-art methods. However, SAM is imperfect or even fails when situations occur, such as weak boundaries, low-contrast, and smaller and irregular shapes. For most situations, the subpar segmentation performance of SAM is not sufficient and satisfying for medical image segmentation tasks. The other line of research Ma & Wang (2023); Wu et al. (2023); Li et al. (2023); Gong et al. (2023) focuses on how to better adapt SAM to medical image segmentation tasks. The requirements of prompts are a difficult problem to deal with. Most of the papers abandoned and designed the prompt encoder or mask decoder to avoid the requirements of prompts. But this way is not advisable since it would destroy the consistent system of SAM and abandon the strong abilities of the prompt encoder and mask decoder, which are trained via large-scale datasets. Therefore, we propose a self-prompt SAM to generate auxiliary masks via the image encoder itself with a designed multi-scale prompt generator. In this way, we not only retain the strong capabilities of each component in SAM but also solve the requirements of extra prompts provided.

## 3 THE PROPOSED METHOD

### 3.1 RETHINKING SAM

In this section, we will introduce the Segment Anything Model (SAM) and analyze existing issues while adapting SAM to medical image segmentation. SAM, as shown in Figure 1(a), is the first prompt-driven foundation model for natural image segmentation, which is trained on the large-scale SA-1B dataset of 1B masks and 11M images, which enables the model with strong zero-shot generalization. The SAM architecture consists of three main components, the image encoder that employs the Vision Transformer as the backbone to extract image features, the prompt encoder that embeds various types of prompts including points, boxes, or texts, and the lightweight mask decoder to generate masks based on the image embedding, prompt embedding, image positional embedding, and output token. We designed our adaptation methods based on two criteria: **(i)** keeping all structures, freezing all weights, and only adding learnable blocks into SAM to perform adaptation since we want to retain all the zero-shot capabilities of SAM, and **(ii)** adapting the model working on from 2D natural image segmentation to 3D medical image segmentation, such as varied modalities and extra depth information. The following will explore the issues while adapting SAM to medical image segmentation.

**Interpolation resize for natural RGB images.** SAM works on natural images which have 3 channels for RGB while medical images have varied modalities as channels. In order to solve the issue

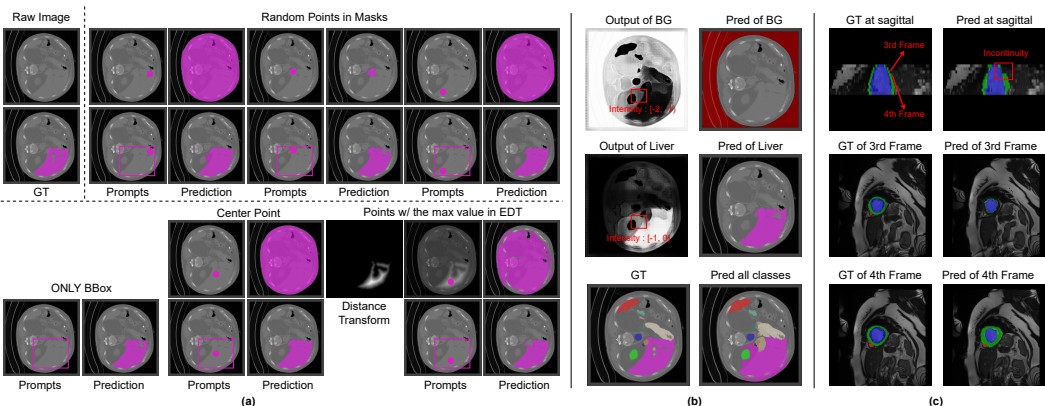

Figure 3: (a) Experiments for different methods to select point. (b) The intensity distribution and prediction of the output of SAM. (c) Incontinuity at the depth dimension.

of how to adapt varied modalities to RGB channels, we adopt a sequence of convolutional layers at the very beginning of the image encoder, which can learn the adaption during finetuning. Moreover, SAM directly upsamples images with any size to a size of $1024 \times 1024$ as the input of the image encoder. For pixel-level tasks, medical segmentation tasks are sensitive to the intensity of each pixel, especially the boundaries of objects. If we directly upsample images to $1024 \times 1024$, it is inevitable to lose information and produce ambiguity. Therefore, we designed an invert-bottleneck architecture built via a sequence of convolutional and transposed convolutional layers to learn the adaption from the varied modalities with any size to 3 channels with a size of $1024 \times 1024$, named MAdapter.

**The requirements of extra prompts.** Since SAM needs to provide prompts, such as points or boxes, when specific objects need to be segmented, many works of SAM-based medical image segmentation need to provide manual prompts during training and inference. It is not always feasible since we can not expect all users to have medical knowledge to provide proper prompts. The rest of the works for adapting SAM to medical image segmentation usually abandon the prompt encoder or mask decoder to avoid the requirements. However, it is not advisable since it would destroy the consistent system of SAM. Meanwhile, it is not wise to abandon the strong prompt encoder and mask decoder, which are trained via large-scale datasets and lots of resources. Therefore, we propose a multi-scale prompt generator (MSPGenerator) utilizing the different levels of feature maps extracted from the image encoder as the input to generate auxiliary masks. Then, points and bounding boxes can be produced based on the auxiliary masks.

**The prompt way.** After we obtain auxiliary masks that can produce points and bounding boxes, we should consider which combination of points, boxes, and masks is the best way for medical image segmentation. Therefore, we conduct experiments that use the ground truth to generate points, bounding boxes, and masks for each class as candidate prompts to find the best combination shown in Figure 1. The experiments demonstrate the prompts of points and points with masks fail since SAM almost segments the entire chest as the liver class (the purple region). The failed reason is the liver region in raw images has weak boundaries and is similar to other regions. When involving bounding boxes, each class can be located in the corrected region though there are errors. The best performance is the prompt of the combination of points, bounding boxes, and masks. Therefore, we chose the combination of points, bounding boxes, and masks as our model's prompt. However, the selection of points can bring enormous differences in performance. The criteria is that the selected point should be as representative of a specific object as possible and inside the mask. It means the point should be as central as possible in the masks. In other words, the points farthest from the boundaries should be selected as the point prompt. Figure 3(a) shows the results of different points with or without bounding boxes. There are enormous differences in performance if randomly selecting points in masks. The performance of selecting the central point of bounding boxes is not the best and the central point is not always located on masks, such as the Myo class (green region) in Figure 3(c). When we utilize the Euclidean distance transform to calculate the distance from boundaries for each pixel and obtain the candidate points farthest from the boundaries as the point prompt, the performance is the best, which is shown in the right-bottom of Figure 3(a).

**Adapting the binary segmentation to multi-scale segmentation.** The original SAM segments all possible objects and does not classify each object belonging to which class, which is more like

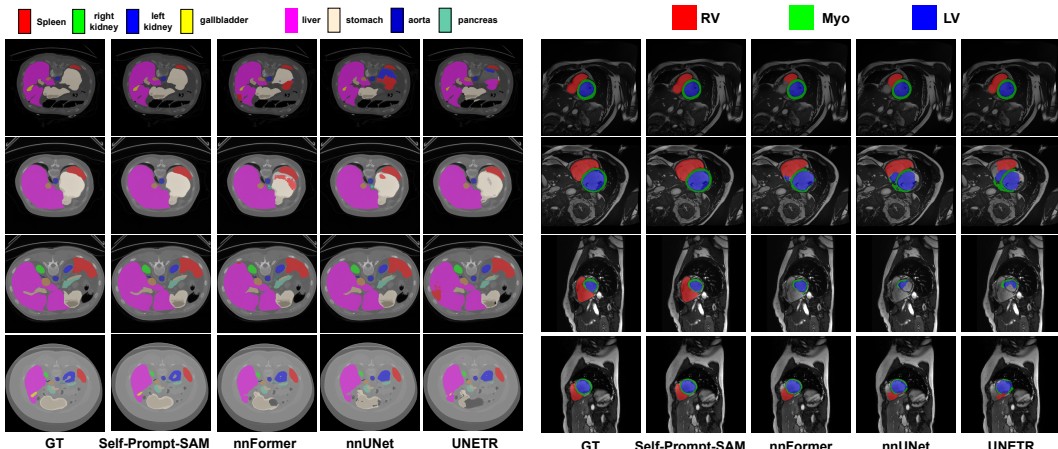

Figure 4: Qualitative comparison on the Synapse and ACDC dataset.

for instance segmentation methods. Our expectation of SAM is to predict each class given raw images as input, like traditional medical image segmentation. When we obtain all outputs for the total classes, each output for a certain class has a different distribution from other outputs, since each output is generated by a specific prompt and trained by a sigmoid function. Therefore, it will obtain very bad results if we directly use a softmax function for all output, which is shown in Figure 3(b). In Figure 3(b), we show two different outputs of SAM for the background and liver class. When we individually consider each class, both of the areas of the red boxes are not considered to belong to its class since the intensities of all pixels are in the range of [-2, -1) and [-1, 0) for background class and liver class, respectively. As long as the intensity of any pixel is less than 0, the pixel will not be considered to belong to the class. However, the area of the red box is considered as the liver class when we adopt a softmax function for all outputs. Therefore, in order to adapt the difference and classify each pixel to only one class, we also designed an invert-bottleneck architecture that consists of two convolutional layers to adapt binary segmentation to multi-class segmentation, named MC=Adapter.

**Adapting with the extra depth dimension.** Since most medical images have an extra depth dimension compared to 2D natural images, it is inevitable to lose 3D spatial information and cause spatial incontinuity at the depth dimension if we directly apply SAM for a sequence of 2D frames shown in Figure 3(c). In Figure 3(c), there exists incontinuity between the 3rd frame and the 4th frame at the depth dimension when we utilize 2D SAM without any operation of the depth information. In order to solve the issue of the incontinuity at depth dimension, we involve learnable depth positional embeddings and modify the original adapter Houlsby et al. (2019) with the ability to explore depth information. i) SAM includes two positional embeddings at the very beginning of the image encoder and the prompt encoder, which we can insert a learnable depth positional embedding, respectively. ii) Inspired by AIM Yang et al. (2023), we introduce an adapter after the multi-head self-attention and in parallel to the MLP layer for each transformer block. We modified the original adapter by adding an invert-bottleneck architecture that consists of two FC layers and an activation layer processing on depth dimension in the middle of the original adapter with a skip connection to learn extra depth information. In this way, our model can learn the extra depth information.

### 3.2 SELF-PROMPT-SAM

In this section, we introduce the whole pipeline of Self-Prompt-SAM for medical image segmentation shown in Figure 2. We retain all structures of SAM and only add designed blocks into SAM.

Given images $X \in \mathbb{R}^{B \times M \times D \times imgH \times imgW}$ with a batch size of $B$, $M$ number of modalities, and a spatial resolution of $D \times imgH \times imgW$. Our goal is to predict the corresponding pixel-wise segmentation with size $B \times N \times D \times imgH \times imgW$, where $N$ is the number of classes of a segmentation task. Firstly, the given images will be resized and reshaped to $BD \times 3 \times 1024 \times 1024$ by our designed MAdapter. Then, the resized images will be fed to the image encoder which consists of a patch embed block, a positional embedding block that adds a learnable depth positional embedding block to learn extra depth information, and a series of transformer blocks that we insert our designed DFusedAdapter after the multi-head self-attention and in parallel to the MLP layer to adapt SAM to

| Method | DSC | Aotra ↑ | Gallbladder ↑ | Kidney(L) ↑ | Kidney(R) ↑ | Liver ↑ | Pancreas ↑ | Spleen ↑ | Stomach ↑ |
|---|---|---|---|---|---|---|---|---|---|
| VNet Milletari et al. (2016) | 68.81 | 75.34 | 51.87 | 77.10 | 80.75 | 87.84 | 40.04 | 80.56 | 56.98 |
| DARR Fu et al. (2020) | 69.77 | 74.74 | 53.77 | 72.31 | 73.24 | 94.08 | 54.18 | 89.90 | 45.96 |
| R50-U-Net Ronneberger et al. (2015) | 74.68 | 87.74 | 63.66 | 80.60 | 78.19 | 93.74 | 56.90 | 85.87 | 74.16 |
| U-Net Ronneberger et al. (2015) | 74.99 | 83.17 | 58.74 | 80.40 | 73.36 | 93.13 | 45.43 | 83.90 | 66.59 |
| R50-AttnUNet Schlemper et al. (2019) | 75.57 | 55.92 | 63.91 | 79.20 | 72.71 | 93.56 | 49.37 | 87.19 | 74.95 |
| VIT-CUP Dosovitskiy et al. (2020) | 67.86 | 70.19 | 45.10 | 74.70 | 67.40 | 91.32 | 42.00 | 81.75 | 70.44 |
| R50-VIT-CUP Dosovitskiy et al. (2020) | 71.29 | 73.73 | 55.13 | 75.80 | 72.20 | 91.51 | 45.99 | 81.99 | 73.95 |
| TransUNet Chen et al. (2021) | 77.48 | 87.23 | 63.16 | 81.87 | 77.02 | 94.08 | 55.86 | 85.08 | 75.62 |
| SwinUNet Cao et al. (2021) | 79.13 | 85.47 | 66.53 | 83.28 | 79.61 | 94.29 | 56.58 | 90.66 | 76.6 |
| TransClaw-U-Net Chang et al. (2021) | 78.09 | 85.87 | 61.38 | 84.83 | 79.36 | 94.28 | 57.65 | 87.74 | 73.55 |
| LeVit-UNet-384s Xu et al. (2021) | 78.53 | 87.33 | 62.23 | 84.61 | 80.25 | 93.11 | 59.07 | 88.86 | 72.76 |
| WAD Li et al. (2021) | 80.30 | 87.73 | 69.93 | 83.95 | 79.78 | 93.95 | 61.02 | 88.86 | 77.16 |
| UNETR Hatamizadeh et al. (2022) | 79.56 | 89.99 | 60.56 | 85.66 | 84.80 | 94.46 | 59.25 | 87.81 | 73.99 |
| nnUNet Isensee et al. (2019) | 86.21 | 92.39 | **71.71** | 86.07 | **91.46** | 95.84 | 82.92 | 90.31 | 79.01 |
| nnFormer Zhou et al. (2021) | 86.57 | **92.40** | 70.17 | **86.57** | 86.25 | 96.84 | **83.35** | 90.51 | 86.83 |
| Self-Prompt-SAM (Ours) | **86.74** | 91.99 | 69.95 | 85.65 | 85.40 | **97.39** | 79.18 | **94.38** | **89.94** |

Table 1: Quantitative results on Synapse dataset (DSC in %).

medical image segmentation and learn extra depth information. After the image encoder, the image embedding is obtained. Meanwhile, we extract feature maps from the image encoder every $n$ time with the feature map of the input of transformer blocks with a size of $BD \times H \times W \times C$.

$$n = (\text{ViT\_Depth} \times H)/imgH, \text{or}, n = (\text{ViT\_Depth} \times W)/imgW \tag{1}$$

All extracted feature maps are fed to our designed MSPGenerator to generate auxiliary masks for each class. The MSPGenerator is a hierarchical structure built by convolutional and transpose convolution layers. Starting with the deepest feature map, it is gradually upsampled to 2x the size and then concatenates with shallower feature maps. Finally, we obtain the auxiliary masks with the same size as the final segmentation with size $B \times N \times D \times imgH \times imgW$. In order to improve performance, alleviate the gradient vanishing, and converge quickly, we involve deep supervision in the MSPGenerator by adding supervision loss at different levels. We utilize the auxiliary masks to generate a point, a bounding box, and a mask for each class. For selecting points, We utilize the Euclidean distance transform to calculate the distance from boundaries for each pixel and obtain the candidate points farthest from the boundaries as the point prompt.

Next, the point, bounding box, and mask prompts are fed into the prompt encoder to generate the sparse embedding, dense embedding, and image positional embedding that also adds a learnable depth positional embedding block to learn extra depth information. The three embeddings are fed into the mask decoder that consists of two cross-attention blocks, which we also insert DFusedAdapter to do adaption and learn extra depth information. Finally, we would obtain a series of binary segmentation masks for each class, which are trained via a sigmoid function and each output for a certain class has a different distribution from other outputs. In order to properly process multi-class segmentation, we equip a M(ulti)C(lasses)-Adapter which is an invert-bottleneck architecture that consists of several convolutional layers to adapt binary segmentation to multi-class segmentation. During training, the output of MC=Adapter will be utilized to calculate the loss with ground truth.

# 4 EXPERIMENTS

## 4.1 DATASETS AND EVALUATION METRICS

We use two publicly available datasets, Synapse multiorgan segmentation Landman et al. (2015) and Automatic Cardiac Diagnosis Challenge (ACDC) Bernard et al. (2018). **(i)** Synapse dataset consists of 30 cases of abdominal CT scans. Following the split strategies Chen et al. (2021), we use a random split of 18 training cases and 12 cases for validation. We evaluate the model performance via the average Dice score (DSC) on 8 abdominal organs (aorta, gallbladder, spleen, left kidney, right kidney, liver, pancreas, and stomach). **(ii)** ACDC dataset consists of 100 patients with the cavity of the right ventricle, the myocardium of the left ventricle, and the cavity of the left ventricle to be segmented. The labels involve the right ventricle (RV), myocardium (MYO), and left ventricle (LV). We use a random split of 70 training cases, 10 validation cases, and 20 testing cases. We evaluate the model performance via the average DSC.

## 4.2 IMPLEMENTATION DETAILS

We utilize some data augmentations such as rotation, scaling, Gaussian noise, Gaussian blur, brightness, and contrast adjustment, simulation of low resolution, gamma augmentation, and mirroring.

| Method | Average ↑ | RV ↑ | Myo ↑ | LV ↑ |
|---|---|---|---|---|
| R50-U-Net Ronneberger et al. (2015) | 87.55 | 87.10 | 80.63 | 94.92 |
| VIT-CUP Dosovitskiy et al. (2020) | 81.45 | 81.46 | 70.71 | 92.18 |
| R50-VIT-CUP Dosovitskiy et al. (2020) | 87.57 | 86.07 | 81.88 | 94.75 |
| UNETR Hatamizadeh et al. (2022) | 88.61 | 85.29 | 86.52 | 94.02 |
| TransUNet Chen et al. (2021) | 89.71 | 88.86 | 84.54 | 95.73 |
| SwinUNet Cao et al. (2021) | 90.00 | 88.55 | 85.62 | 95.83 |
| LeViT-UNet-384s Xu et al. (2021) | 90.32 | 89.55 | 87.64 | 93.76 |
| nnUNet Isensee et al. (2019) | 91.61 | 90.24 | 89.24 | 95.36 |
| nnFormer Zhou et al. (2021) | 92.06 | 90.94 | 89.58 | 95.65 |
| Self-Prompt-SAM (Ours) | **93.26** | **92.20** | **91.22** | **96.36** |

Table 2: Quantitative evaluation with SOTA methods on the ACDC dataset (dice score in %).

We set the initial learning rate to 0.01 and employ a "poly" decay strategy in Eq. 2.

$$lr(e) = init\_lr \times (1 - \frac{e}{\text{MAX\_EPOCH}})^{0.9}, \qquad (2)$$

where $e$ means the number of epochs, MAX_EPOCH means the maximum of epochs, set it to 1000 and each epoch includes 250 iterations. We utilize SGD as our optimizer and set the momentum to 0.99. The weighted decay is set to 3e-5. We utilize both cross-entropy loss and dice loss by simply summing them up as the loss function. We utilize instance normalization as our normalization layer. Since we expect relatively good auxiliary masks to finetune the mask decoder, only the deep supervision loss of MSPGenerator is trained in the first two hundred epochs, and after two hundred epochs, our model combines the deep supervision loss of MSPGenerator and the loss of MC=Adapter at the end of the mask decoder. All experiments are conducted using single NVIDIA RTX A6000 GPUs with 40GB memory.

## 4.3 COMPARISON WITH STATE-OF-THE-ART METHODS

**Results on Synapse Dataset.** In Table 1, we present the quantitative experimental results on the Synapse dataset compared to several leading convolution-based methods (*i.e.*, VNet Ronneberger et al. (2015) and nnUNet Isensee et al. (2019)) and transformer-based methods (*i.e.*, TransUNet Chen et al. (2021), SwinUNet Cao et al. (2021), LeViT-UNet-384s Xu et al. (2021)), and nnFormer Zhou et al. (2021). We observe that our proposed Self-Prompt-SAM outperforms all existing methods and achieves a new state-of-the-art performance. Meanwhile, our model predicts well in the large-size 'Liver' label, 'Spleen' label, and 'Stomach' label due to our proposed DFusedAdapter that can learn more 3D spatial information and adapt 2D SAM to medical image segmentation. In Figure 4, we illustrate the qualitative results compared with representative methods. These results also demonstrate that our Self-Prompt-SAM can predict more accurately the 'Liver' label, 'Spleen' label, and 'Stomach' label. Therefore, the results demonstrate the effectiveness of our method.

**Results on ACDC Dataset.** In Table 2, we provide the quantitative experimental results on ACDC dataset. Specifically, we compare the proposed Self-Prompt-SAM with several leading convolution-based methods (*i.e.*, R50-U-Net Ronneberger et al. (2015) and nnUNet Isensee et al. (2019)) and transformer-based methods (*i.e.*, TransUNet Chen et al. (2021), SwinUNet Cao et al. (2021), and LeViT-UNet-384s Xu et al. (2021)). The results show that the proposed Self-Prompt-SAM outperforms various state-of-the-art approaches, surpassing nnFormer by 1.2%, 2.3%, 1.7%, and 0.7% in DSC, RV dice, Myo dice, and LV dice. In Figure 4, we provide the qualitative results compared with several state-of-the-art methods. As shown in Figure 4, our Self-Prompt-SAM model can predict more accurately on all labels. Meanwhile, the results demonstrate the effectiveness of our method since our proposed modules can properly solve the drawbacks of SAM when adapting to medical image segmentation.

## 4.4 ABLATION STUDY

We also evaluate the effectiveness of several variants of network architectures.

*(A) Baseline Models.* The proposed Self-Prompt-SAM has 9 baselines (*i.e.*, S1, S2, S3, S4, S5, S6, S7, S8, S9) as shown in Table 3. All baselines adopt the whole structures of SAM and only add blocks. (i) S1 adopts a series of stacked CNNs for the prompt generator combined with the image encoder. (ii) S2 utilizes our proposed multi-scale prompt generator (MSPGenerator) to combine with the image encoder to generate prompts. (iii) S3 adds the vanilla adapter Houlsby et al. (2019) with each transformer block in the image encoder and mask decoder based on S2. (iv) S4

| | Method | DSC ↑ |
|---|---|---|
| S1 | SAM + stacked CNNs prompt generator | 79.57 |
| S2 | SAM + MSPGenerator | 82.20 |
| S3 | SAM + MSPGenerator + vAdapter | 90.08 |
| S4 | SAM + MSPGenerator + vAdapter w/ Depth MLPs before vApdater | 91.45 |
| S5 | SAM + MSPGenerator + vAdapter w/ Depth MLPs after vApdater | 91.52 |
| S6 | SAM + MSPGenerator + DFusedAdapter | 91.73 |
| S7 | SAM + MSPGenerator + DFusedAdapter + DPosEmbed | 91.88 |
| S8 | SAM + MSPGenerator + DFusedAdapter + DPosEmbed + MAdapter | 92.20 |
| S9 | Our Full Model (S6 + MC=Adapter) | **93.26** |

Table 3: The ablation studies of the proposed method on the ACDC dataset. The MSPGenerator means the multi-scale prompt generator. The vAdapter means the vanilla adapter. The DPosEmbed means the depth positional embedding.

adds the modified adapter by inserting the invert-bottleneck depth MLPs before the adapter with a skip connection based on S2. (v) S5 adds the modified adapter by inserting the invert-bottleneck depth MLPs with a skip connection after the adapter based on S2. (vi) S6 adds our DFusedAdapter with each transformer block in the image encoder and mask decoder based on S2. (vii) S7 adds depth positional embedding blocks (DPosEmbed) in the image encoder and prompt encoder based on S6. (viII) S8 adds the modalities adapter (MAdapter) before the image encoder based on S7. (ix) S9 is our full model, named Self-Prompt-SAM, illustrated in Figure 2. S9 adds multi-classes adapter (MC=Adapter) at the end of SAM to adapt binary segmentation to multi-classes segmentation based on S8. The results of the ablation study are shown in Table 3.

*(B) Effect of MSPGenerator.* When we use an MSPGenerator to generate auxiliary masks by involving different levels of feature maps from the image encoder and gradually upsampling the spatial dimension, the average DSC of S2 improves by 2.7% compared with S1. The result confirms the effectiveness of the proposed MSPGenerator.

*(C) Effect of DFusedAdapter.* To insert adapters into each transformer block, the performance of S3 has a huge improvement of 8% compared to S2, which demonstrates that using adapters to finetune SAM to medical image segmentation is feasible. Meanwhile, we found the performance of S4 and S5 are very close when we insert the depth MLPs with a skip connection before or after the vanilla adapter. But the DFuserAdapter that we insert the depth MLPs with a skip connection in the middle of the vanilla adapter achieves the best performance compared to S4 and S5. Moreover, S6 improves by 1.7% compared to S3. The result confirms the effectiveness of the proposed DFusedAdapter.

*(D) Effect of DPosEmbed blocks.* The performance of S7 improves by more than 0.1% compared to S6, which demonstrates the effectiveness of the DPosEmbed blocks.

*(E) Effect of MAdapter.* When we adopt the MAdapter before the image encoder, the average DSC of S8 improves by 0.4% compared to S7, which confirms the benefits of the MAdapter.

*(F) Effect of MC=Adapter.* S9 is our full model, Self-Prompt-SAM, utilizing an MC=Adapter at the end of SAM to adapt binary segmentation to multi-classes segmentation based on S6 as shown in Figure 2. Compared to S8, our model brings 1% improvements. Therefore, the results demonstrate the effectiveness of our proposed Self-Prompt-SAM.

## 5 CONCLUSION

In this paper, we propose a self-prompt SAM adaptation framework for medical image segmentation, named Self-Prompt-SAM, which adapts pre-trained SAM models worked on from 2D natural images to 3D medical images without any prompts provided. By designing a multi-scale prompt generator (MSPGenerator) combined with the image encoder in SAM to generate auxiliary masks, our model can generate prompts by itself. The auxiliary masks would be used to generate bounding boxes as box prompts and utilize Distance Transform to select the points farthest from the boundary as point prompts. In order to learn extra depth information, we designed a 3D depth-fused adapter (DfusedAdapter) that adds an invert-bottleneck architecture that consists of two FC layers and an activation layer processing on depth dimension in the middle of the original adapter with a skip connection and injected a series of DFusedAdapter into each transformer block in the image encoder and mask decoder to enable pre-trained 2D SAM models to extract 3D information and adapt to medical images. Extensive experiments demonstrate that our method outperforms existing state-of-the-art approaches on two challenging public ACDC Bernard et al. (2018) and Synapse Landman et al. (2015) medical image segmentation datasets.

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
