# OpenReview forum: "Self-Prompt SAM: Automatic Prompt SAM Adaptation for Medical Image Segmentation"
_ICLR.cc/2024/Conference — ICLR 2024 Conference Withdrawn Submission_

### Official Review · Reviewer_4QxB · 2023-10-28

**Soundness:** 3 good
**Presentation:** 1 poor
**Contribution:** 2 fair
**Rating:** 3
**Confidence:** 5

**Summary:**

SAM is a pre-trained foundation model for image segmentation. However, prior arts have shown that SAM performs poorly on medical images due to the semantic gap between its training images and the medical domain. This paper tackles the poor performance of SAM on medical images. It concludes that the segmentation results can be greatly improved with a proper combination of bounding box prompts and point prompts. Based on this, automatic prompts are generated from the pre-trained encoder of SAM for medical queries. In addition, different adaptors are designed for tailoring SAM from 2D to 3D format. Experiments on two different medical datasets show the effectiveness of the proposed method over prior arts.

**Strengths:**

(1) The target problem with SAM is very important. Properly addressing it would significantly improve medical image segmentation in the field.

(2) The motivations of the technical innovations are clearly presented. The paper is easy to follow and understand.

(3) Extensive ablation studies are conducted to demonstrate the effectiveness of the adaptation.

**Weaknesses:**

(1)  A significant drawback of this paper lies in the insufficient methodological exposition. The specifics regarding the implementation and training procedures of the proposed adaptor modules remain unclear. For instance, how is each of the proposed adaptor modules implemented and trained? What are the loss functions? Are all of the adaptors trained jointly? This lack of comprehensive information hinders the reproducibility of the research.

(2) The figures presented in the paper are somewhat perplexing. They require more elaborate descriptions and additional contextual explanations.

(3) Minor issues of typos and gramma errors.

**Questions:**

Please refer to the weakness section.

---

### Official Review · Reviewer_NtPf · 2023-10-31

**Soundness:** 2 fair
**Presentation:** 2 fair
**Contribution:** 2 fair
**Rating:** 5
**Confidence:** 4

**Summary:**

The paper endeavors to adapt an existing model to a three-dimensional context without relying on external prompts. It introduces a multi-scale strategy as the cornerstone of its method for a particular downstream task. In this model, a multi-scale prompt generator is seamlessly integrated with the image encoder in SAM, enabling it to autonomously generate auxiliary masks. These masks are then used to form bounding boxes that act as box prompts. Furthermore, the model employs the Distance Transform method to identify points that are farthest from the boundary, which are subsequently utilized as point prompts.

**Strengths:**

The paper showcases advancements in prompt generation and model adaptation, particularly in 3D contexts and medical imaging applications.

The proposed generator leverages multiple levels of information from the feature map, enhancing its effectiveness and versatility in processing complex data.

It demonstrates an intriguing approach by combining box points and masks to generate prompts. This hybrid method potentially offers a more robust way of prompt generation compared to traditional methods.

The method's efficacy is thoroughly tested and validated on two different datasets.

A remarkable innovation is the development of a 3D depth-fused adapter. This enables pre-trained 2D SAM models to extract and interpret 3D information, making it useful in adapting to medical imaging, a field where 3D data is prevalent

**Weaknesses:**

A major part of the method is introducing a self-prompted approach within the SAM framework, primarily focused on segmentation without external prompts. However, the most compelling aspect of SAM is its zero-shot learning capability across various datasets. By entirely removing manual prompts, the framework diverges from its zero-shot learning roots, shifting more towards a transfer learning approach. This change might be seen as a step back from the innovative aspects of SAM, as it no longer operates under the originally intended zero-shot learning setting.

The DfusedAdapter appears to be a critical element, especially evident in Table 3 of the paper. However, its presentation in the methodology section is somewhat unclear or insufficiently detailed. This lack of clarity could hinder the understanding and reproducibility of the research.

The concept of using an adapter or fine-tuning the SAM model to achieve effective segmentation without manual prompts is not entirely new. Similar approaches have been previously proposed in related works. Therefore, the paper might yield insufficient innovation in this specific aspect of the research.

**Questions:**

Please see my comments in "weakness".

---

### Official Review · Reviewer_VLNv · 2023-10-31

**Soundness:** 2 fair
**Presentation:** 3 good
**Contribution:** 2 fair
**Rating:** 3
**Confidence:** 4

**Summary:**

This paper proposes a  self-prompt SAM adaptation framework for medical image segmentation,
termed Self-Prompt-SAM. A multi-scale prompt generator is combined with the image encoder in
SAM to generate auxiliary masks. The auxiliary masks are used  to generate bounding boxes as box prompts  and Distance Transform is utilized to select the points farthest from the boundary as point prompts.

Results are shown on two benchmark datasets.

**Strengths:**

Detailed complex architecture of the proposed Self-Prompt-SAM, as in in figure 2.

3D transformer block designed; utilizing the 2D SAM with designed adapters to equip
spatiotemporal reasoning capability for 3D medical image segmentation.

Good that implementation requires only one GPU with large memory, as reported.

**Weaknesses:**

Only one equation as analytics; That too is comes without continuity of text - just in between two paras.
(although purpose in clear).

No algorithm for learning (new of modified) proposed.

Very difficult to infer goodness of the results in figs. 3 & 4, as details are too minor for the naive (non-med experts) eye . Should have zoomed onto certain regions/parts of images to exhibit the better detection of proposed method.

How is the DT feature helping the cause of better detection - is not clear.

Also how the 3D data is analyzed and corresponding results obtained wrt GT - is not vivid in presentation.

**Questions:**

What is the performance in too much of clutter
or detection of small objects in medical images ?

Why are failure cases not discussed?

Will the code be released ?

---

### Official Review · Reviewer_wtde · 2023-10-31

**Soundness:** 2 fair
**Presentation:** 2 fair
**Contribution:** 2 fair
**Rating:** 3
**Confidence:** 4

**Summary:**

In this manuscript the authors propose to adapt the Segment Anything Model (SAM) to 3D prompt-free medical image segmentation tasks, using 1. adapters; 2. a prompt generator network which produces prompts (essentially a pre-segmentation stage). The proposed approach is evaluated on abdominal CT and cardiac MRI segmentations. It demonstrates improved segmentation performance compared to some pre-SAM approaches that do not involve large-scale pre-training.

**Strengths:**

The topic of leveraging foundation models for domain-specific applications is of practical value, despite the surge of similar works.

Improved segmentation accuracies are shown on two public medical image segmentation datasets, compared with pre-SAM approaches.

The idea of improving automation by removing the requirement on manual prompting is of great clinical value.

**Weaknesses:**

Similarity to existing works: The readers may argue that merely combining 1. adapters (has been studied by [1,2,3]), 2. modified prompt encoder/prompting mechanism (studied by [2,5]) for auto-prompting, and 3. operations along the depth dimension ([1,2,3]), may not meet the high standard of ICLR. ICLR encourages theoretical insights into the fundamentals of the problems, and novel ideas that can inspire related works. Despite the claimed further automation compared with manually-prompted SAM's, [1,4,5] can also be directly trained and tested in without manual prompts, meeting the same level of convenience as the proposed work.

The writing style needs to be improved:

1. The manuscript is poorly structured. It is understandable that the authors went through a chain of thinking; but this mixed and convoluted narration, without organization, may severely mislead readers, especially for the introductory paragraph. The authors are expected to make their language concise: straightforward to the key arguments, and then talk about supporting evidences and/or reasoning behind. Fixing this structural flaw may require substantial amount of efforts that is beyond what can be expected for a rebuttal.

2. Arguments with little clarity/grammatic errors:
a) “Therefore, adapting SAM to medical image segmentation tasks is the main direction by modifying the structure of SAM”, what does this mean?
b) “We not only maximize the utilization of the capabilities of SAM but also adapt SAM to medical image segmentation, which is a trade-off.” why?
c) “When we obtain all outputs for the total classes, each output for a certain class has a different distribution from other outputs, since each output is generated by a specific prompt and trained by a sigmoid function. Therefore, it will obtain very bad results if we directly use a softmax function for all output, which is shown in Figure 3(b).” What does this mean?

3. Unsupported claims:
d) “Most of the papers abandoned and designed the prompt encoder or mask decoder to avoid the requirements of prompts. But this way is not advisable since it would destroy the consistent system of SAM and abandon the strong abilities of the prompt encoder and mask decoder, which are trained via large-scale datasets.” Given that you are tuning on new data anyway, discarding the prompt encoder may not necessarily be a drawback unless there are substantial evidences.
e) “SAM, as shown in Figure 1(a), is the first prompt-driven foundation model for natural image segmentation.” Wrong. Works such as CLIPSeg come out much earlier.

4. Sloppy mathematical notations in Sec. 3.2. The authors are encouraged to use superscripts and subscripts properly.

5. Fig 2. is complicated and confusing.

The experiments are overly simplistic: segmenting abdominal CT on Synapse dataset and cardiac MRI on ACDC dataset, in late 2023, are overly simplistic, and they are commonly agreed to be solved problems. The authors are expected to demonstrate how SAM-based models shed light on unsolved problems, e.g., dealing with more challenging segmentation targets and/or using less training data. Also, the authors are expected to make comparisons with SAM-derived, manual-prompt-free approaches, e.g., [1,2,4]


[1] Medical sam adapter: Adapting segment anything model for medical image segmentation

[2] Auto-Prompting SAM for Mobile Friendly 3D Medical Image Segmentation

[3] 3DSAM-adapter: Holistic Adaptation of SAM from 2D to 3D for Promptable Medical Image Segmentation

[4] How to Efficiently Adapt Large Segmentation Model(SAM) to Medical Images

[5] AutoSAM: Adapting SAM to Medical Images by Overloading the Prompt Encoder

**Questions:**

The authors are encouraged to improve the clarity of writing, e.g., becoming more straightforward to the key arguments, instead of reaching an conclusion with convoluted chain of arguments; using more concise language; avoiding groundless claims.

The authors are encouraged to improve the clarity of the illustrations, especially for Fig. 2. It would be more readable if a high-level overview of the approach can be first presented, before diving into endless details.

**Details Of Ethics Concerns:**

The authors need to be aware of potential biases inherent in deep learning models for medical imaging applications (not specific to this work).